# Emerging Parvoviruses in Domestic Cats

**DOI:** 10.3390/v13061077

**Published:** 2021-06-04

**Authors:** Paolo Capozza, Vito Martella, Canio Buonavoglia, Nicola Decaro

**Affiliations:** Department of Veterinary Medicine, University of Bari Aldo Moro, 70010 Valenzano, Italy; paolo.capozza@uniba.it (P.C.); vito.martella@uniba.it (V.M.); canio.buonavoglia@uniba.it (C.B.)

**Keywords:** cat, emerging parvoviruses, protoparvoviruses, bocaparvoviruses, chaphamaparvoviruses

## Abstract

Parvovirus infections in cats have been well known for around 100 years. Recently, the use of molecular assays and metagenomic approaches for virus discovery and characterization has led to the detection of novel parvovirus lineages and/or species infecting the feline host. However, the involvement of emerging parvoviruses in the onset of gastroenteritis or other feline diseases is still uncertain.

## 1. Introduction

*Parvoviridae* is a large and remarkably diverse family of small (22–30 nm in diameter), non-enveloped, icosahedral viruses. The parvoviral genome is a positive-sense single-stranded (ss) DNA (4.5–5.5 kb), with complex hairpin-like structures at the 5′ and 3′ ends [1,2,3]. The coding region of the genome contains two major expression cassettes, with open reading frames (ORFs) on the left-hand side giving rise to non-structural (NS) proteins (*ORF1*), whereas mRNA populations responsible for translating structural proteins (VPs) are transcribed from the right-hand cassette (*ORF2*) [1,2,4,5].

In recent years, using molecular assays and metagenomic approaches for virus discovery and characterization, different research groups have detected novel lineages and species of parvoviruses in cats, leading to a change in the classification of the family *Parvoviridae* [1,2,4,6]. Indeed, according to the classification criteria of the International Committee on Taxonomy of Viruses (ICTV), the *Parvoviridae* family is currently divided into three subfamilies: *Parvovirinae* and *Densovirinae*, which infect vertebrates and arthropods, respectively, and the new subfamily *Hamaparvovirinae*, which infects both [1,2,3,4,6] (Table 1).

Parvoviruses have a large host spectrum, spanning from invertebrates to mammals [2,7,8]. Since the first identification in 1928 from the fecal samples of cats with gastroenteritis [9,10], feline panleukopenia virus (FPV), currently included in the species *Carnivore Protoparvovirus 1* of the genus *Protoparvovirus* [1], has been causing the most important parvoviral disease in cats. All members of the family *Felidae* are probably susceptible to infection with FPV, which occurs worldwide. Other carnivores of the families *Viverridae, Procyonidae*, and *Mustelidae* also are susceptible to infection, although only a smaller number of hosts have been observed to suffer clinical disease, including raccoon (*Procyon lotor*), mink (genera *Mustela* and *Neovison*), and coatimundi (genus *Nasua*) [7]. Most wild carnivores are also susceptible to the closely related *Carnivore Protoparvovirus 1*, canine parvovirus (CPV) [11]. In cats, FPV infection causes feline panleukopenia (FPL), a highly contagious, often fatal disease, characterized by acute severe enteritis, dehydration and sepsis due to lymphoid depletion and pancytopenia [12]. Infection spreads rapidly, especially in cells with high mitotic activity, such as bone marrow, lymphoid tissues, and intestinal crypt cells.

Anorexia, vomiting, diarrhea, neutropenia, and lymphopenia are common in clinically affected cases. Kittens are most severely affected. In utero or neonatal infection can result in cerebellar hypoplasia. Depending on the severity of the clinical signs, mortality ranges from 25% to 100%. FPL is now diagnosed infrequently by veterinarians in several countries, presumably as a consequence of widespread vaccine use [12,20]. For example, in Australia, there had been no outbreaks of FPL reported even in shelters for over 30 years [12], but multiple outbreaks occurred in eastern Australia between 2014 and 2018 [21]. Infection rates remain high in some unvaccinated cat populations, and the disease occasionally is seen in vaccinated, pedigreed kittens that have been exposed to a high-titer virus challenge [12,20].

In agreement with current ICTV guidelines, parvoviruses are considered members of the same species if their NS1 proteins share more than 85% amino acid (aa) sequence identity. They can be classified into the same taxon if their protein sequences cluster as a robust monophyletic lineage based on their complete NS1 protein sequence at the subfamily level and on their SF3 helicase domains at the family level. Additionally, NS1 proteins of members of the same genus should share at least 35–40% aa sequence identity, with a coverage of >80% between any two members. Failing the sequence identity-based criteria, common genus affiliation can also be justified based on a similar genome organization, i.e., presence or absence of certain auxiliary-protein-encoding genes, genome length, and/or transcription strategy [1,2,4,6].

There is limited information on the epidemiology and genetic heterogeneity of these new parvoviruses, and it is unclear whether these viruses could play a role as enteric pathogens in cats and what is their impact on feline health. The aim of this review is to provide an update on emerging feline parvoviruses that have most recently been identified in association (or not) with enteric signs.

## 2. Protoparvoviruses

The species *Carnivore protoparvovirus*
*1*, within genus *Protoparvovirus* (Table 1), includes genetically and antigenically related viruses such as FPV, CPV, and parvoviruses of wild animals, all causing clinically important diseases, especially in young animals [1,3,4,6,22,23,24,25,26].

FPV has been known since 1928 [9,10], while CPV emerged as a dog pathogen in the late 1970s, most likely as a host variant of the feline virus [27] and thanks to an unknown putative adaptive host [23,28,29]. Currently, the evolutionary studies and host jumps of protoparvoviruses in carnivores are generating strong scientific interest around the world.

It is known that, among parvoviruses, CPV evolves more rapidly than FPV [30], showing higher rates of nucleotide changes [31,32,33,34]. Indeed, a few years after its onset, the original strain CPV-2 gave way to two antigenic variants, CPV-2a and CPV-2b [35,36]. In 2000, a third variant, CPV-2c, was identified in Italy [37] and found to spread quickly in all continents, with the exception of Australia [29,38,39,40,41,42,43,44,45,46,47,48,49,50]. Although the main host is the dog (*Canis lupus familiaris*) [51], CPV variants are able to infect numerous other carnivores, including cats (*Felis catus*) [24,52,53].

Unlike the original CPV-2, which does not infect cats, its antigenic variants have been widely isolated from the blood [54,55] and feces of cats worldwide, in both natural [13,52,55,56,57,58,59,60,61,62] and experimental [63] infections. CPV was detected in cats for the first time in the late 1980s, when CPV-2a-like strains were isolated from non-symptomatic cats in Japan [13]. Subsequently, CPV was also detected in cats with the identification of 2b strains in the USA [64] and of CPV-2a/2b strains in Germany [52].

In several cases of natural infection, the CPV variants were reported in the feces of healthy cats [13,33,65,66], as a probable outcome of prolonged shedding or subclinical infection [33,55]. A longitudinal study, carried out on cats hosted in two shelters, showed a high prevalence of CPV, in the total absence of clinical signs and with prolonged fecal shedding for up to 6 weeks, suggesting a possible role of cats as reservoirs and asymptomatic carriers for CPV infection [65]. This hypothesis was supported by a more recent study, which detected the DNA of FPV and CPV variants (CPV-2b and 2c) in the white blood cells (WBCs) of asymptomatic cats, despite the presence of specific antibodies against parvoviruses [55].

On the other hand, another study, conducted in Australia in 2018, reported the extremely low prevalence of CPV fecal shedding in asymptomatic shelter-housed cats, concluding that the fecal spread of CPV by healthy cats could not represent a real risk of infection in mixed cat–dog shelters [67].

Many other studies report CPVs as a cause of clinical signs in cats that are indistinguishable from those induced by FPV [24,47,54,59,60,61,62,68], although this virus remains the most common etiological agent of FPL [52,67,68]. CPV variants were also identified in nervous tissues, posing the question of the replicative ability of these viruses in differentiated and not mitotically active cells [53,69]. One of the first cases of natural CPV infection in cats occurred in a 1.5-year-old female cat with typical signs of panleukopenia [56]; another case of a 5-month-old kitten with classic presentation of FPL, characterized by depression, fever, dehydration, and marked leukopenia, was associated to CPV-2c infection [62].

Subsequently, CPV variants were retrieved from cats with FPL-like disease in Italy, Germany, USA, Japan, India, Portugal, and Spain [24,47,52,53,55,56,58,59,60,61,70,71]

Interestingly, as cats are susceptible to both CPV variants and FPV, superinfection and co-infection with multiple *Carnivore protoparvovirus 1* strains may occur, potentially facilitating the recombination and high genetic heterogeneity of these viruses. Indeed, the detection of mixed FPV/CPV infections in cats, associated or not with clinical signs, emphasizes the possible role of cats as a source of new variants of parvoviruses, with concrete problems for the implementation of prophylaxis [55,59,60,61]. Recently, the ability of FPV to replicate in the dog thymus has been demonstrated in experimental infections [72], but the virus has not been detected in the canine intestine so far.

In 2017, in Italy, a new protoparvovirus strain was identified from cats with or without signs of upper respiratory tract disease and from stool samples of diarrheic animals [14] (Table 1). This virus shared more than 99.9% nt sequence identity of VP2 with canine bufavirus (CBuV), previously detected in a litter of five-month-old puppies during an outbreak of canine infectious respiratory disease (CIRD) in Italy [73]. CBuV displayed low aa identity (19.3–51.4%) in the NS1 protein to members of the species *Carnivore protoparvovirus 1*, while the closest relatives to CBuV (47.2–51.4% aa identity in NS1) were protoparvoviruses identified in human and non-human primates, commonly termed bufaviruses (BuVs) [74,75,76]. In agreement with the new ICTV classification criteria, these canine and feline protoparvoviruses could be considered members of a new species, which has been referred to as *Carnivore protoparvovirus 2*, within the genus *Protoparvovirus*, [14,73].

In humans and, more recently, in wild animals (wolves and foxes), BuVs were identified almost exclusively in the enteric tract [77,78]. However, investigations in dogs [73,79], monkeys [76], shrews [80], and sea otters [81] suggest possible extraintestinal and/or systemic BuV infections. In a more recent study conducted in China, CBuVs were detected in sera from dogs with signs of CIRD [79]. Similarly, feline strains appear to be part of the feline respiratory virome, since BuVs have been detected more frequently in respiratory samples than in rectal swabs, with a possible age-related pattern of infection [14].

Accordingly, in domestic carnivore species, a preferential tropism of these viruses for the respiratory tract has been hypothesized, although their identification in the enteric tract of domestic carnivores and, more recently, of wild canids [77] should be further investigated in order to rule out their enteropathogenic role, as well as the possibility that fecal shedding of BuVs represents a strategy of virus persistence in animal populations.

The multi-host nature of viruses belonging to genus *Protoparvirus* and the ability of some of them to induce severe clinical signs are aspects that must be considered in the implementation of individual and collective prophylaxis plans, in order not only to limit the spread of these viruses between individuals of the same species, but also to prevent the transmission of other carnivore species that share the same environment.

## 3. Bocaparvoviruses

Bocaviruses (BoVs) are members of the genus *Bocaparvovirus* (subfamily *Parvovirinae*) that cause disease in various animals and humans, including porcine BoVs [82,83], bovine parvovirus [84], California sea lion BoV [85], bats BoV [86], rabbit BoV [87], rodent BoV [88], pine martens BoV [89], mink BoV [90], canine BoVs [15,91,92], feline BoVs [15,16,17,93], gorilla BoV [94,95], and human BoVs [96,97], thus suggesting a potentially wide host range of these parvoviruses.

BoVS are unique among parvoviruses since they contain an additional ORF (*ORF3*), located between the non-structural (*ORF1*) and structural (*ORF2*) coding regions of their genome, a 5.5-kb ssDNA (Figure 1). *ORF3* encodes for NP1, a highly phosphorylated protein that is not similar to proteins of other parvoviruses and plays a role in RNA processing. NP1 controls the splicing of VP-encoding RNAs and read-through of the proximal polyadenylation [3,98,99,100].

Following ICTV classification criteria, BoVs are classified into twenty-five officially recognized species, of which at least five species have been detected in domestic carnivores (*Carnivore bocaparvovirus 1–5*), while a sixth species (*Carnivore bocaparvovirus 6*) was found in mink [3,4,6,90]. Currently, feline bocaparvoviruses (FBoVs) identified in domestic cats are classified within the species *Carnivore bocaparvovirus 3* to *5* [15,16,17] (Table 1). FBoV DNA was detected for the first time in 2012, using molecular tools, in feces, nasal swabs, urine, kidney, and blood collected from stray cats in Hong Kong [15]. The near-complete genomic sequences (5179–5331 nt) were obtained from two fecal samples (HK797F and HK875F) and from a urine sample (HK797U) of two cats. After sequence analysis, the three FBoV genomes showed an identity of 58.6–59.7% to canine minute virus [15,91].

In agreement with new strict classification criteria, the newly discovered feline parvoviruses have been classified within a species named *Carnivore bocaparvovirus 3* (subspecies FBoV-1) [1,3,4,6]. Subsequently, using a metagenomics approach on nucleic acids of enriched viral particles from the feces of a single healthy cat in Portugal [16], an additional near-complete bocavirus genome was sequenced (strain POR1). Strain POR1 shares an aa identity of 58% for NS1 and of 70% for VP1 with the three FBoVs previously identified in Hong Kong [15,16]. Accordingly, strain POR1 has been classified as a distinct species, *Carnivore bocaparvovirus 4* (FBoV-2) [16]. A third strain of feline bocavirus (FBoV-3) was detected in 2014 in fecal pools collected from 25 cats from a shelter in California using next-generation sequencing (NGS). FBoV-3 has been accepted as a prototype of an additional species, *Carnivore bocaparvovirus 5*, since it shares an aa sequence identity of 68% for NS1 and of 76% for VP1 with the other feline bocavirus strains previously identified [17].

Since then, different genomes of FBoV have been reported in Europe, China, Japan, Thailand, and Canada from the feces of cats with and without clinical signs [16,19,93,101,102,103]. This suggests that different FBoVs circulate in cats in different geographical areas, without any unambiguous correlation between FBoV genetic diversity and biological properties.

FBoV-1 infection could pose a concrete health risk to cats since it has been associated with enteritis [104,105]. Several studies have shown that FBoV-1 is more likely to be detected in cats with diarrhea than in healthy cats [93,102,103,104]. In a molecular survey performed in China, FBoV-1 DNA was found in 7.7% of the feces from cats with severe enteritis, while it was not detected in healthy animals [102]. In another investigation conducted in China, FBoV-1 DNA was evident in 2.8% of cats with severe enteritis, with a statistically significant association between FBoV-1 infection and the presence of diarrhea [103]. In a more recent study carried out in Thailand, three independent outbreaks of hemorrhagic enteritis in household cats associated with FBoV-1 infection were described [93].

Moreover, the detection of FBoV-1 DNA in multiple tissues, including fecal, urine, blood, respiratory, and kidney samples, collected from stray cats in Hong Kong, suggests a wide tissue tropism [15].

On the other hand, whether FBoV-2 and 3 can play a role in the occurrence of gastroenteritis or other feline diseases remains unclear. Indeed, these viruses were repeatedly found in fecal samples of healthy cats [16,17]. In a Japanese study, the FBoV-2 genome was identified in rectal swabs collected from healthy cats (8.4%) and from cats with gastroenteritis (11.32%), without a statistically significant association between viral DNA detection and the presence of clinical signs [101].

In most cases, FBoV DNA was co-detected in cats with other viral pathogens such as FPV [93], fechaviruses [19], rotaviruses, astroviruses, bocaviruses, sakobuviruses, and/or picobirnaviruses [16,22,101], suggesting that FBoV could be considered a common component of the feline fecal virome. Although the role of systemic infection of cats remains undetermined, a pathogenic role of FBoV enteric infection is possible. Synergistic effects of co-infections with other enteric viruses could lead to more severe clinical signs, such as hemorrhagic enteritis [93,102].

Including systematically FBoVs in the diagnostic algorithm of feline viral enteritis, using specific molecular tools, could help us to better understand the enteropathogenic potential of these viruses and the possible correlation between the genetic diversity and the biological proprieties of each different FBoV species.

## 4. Chaphamaparvoviruses

The genus *Chaphamaparvovirus* (ChPV) (subfamily *Hamaparvovirinae*), recently introduced in the family *Parvoviridae,* includes viruses genetically more related to invertebrate-infecting parvoviruses than to members of the subfamily *Parvovirinae.* Future detection and characterization of new viruses related to currently recognized members of this proposed taxon might eventually result in splitting the currently recognized single genus into more genera. Currently, however, clustering of these viruses as a single genus is the only common node characterized by significant topology support by both Bayesian and maximum likelihood-based inference [2,3,4,6].

After the first identification in oropharyngeal swab samples collected from a fruit bat (*Eidolon helvum*) in Ghana [106], ChPV-like viruses have been reported in several additional animal species [2], including dogs and cats [18,19,107,108,109]. The first description of ChPV in domestic carnivores occurred in the USA in 2017, where an NGS approach was used on the feces of two dogs with hemorrhagic diarrhea of unknown etiology [107]. Later, viruses genetically close to the American canine ChPV strains were found in the feces of dogs and cats in China and Italy [18,108,109]. In agreement with ICTV classification criteria, all strains of canine and feline (FeChPV) origin have been segregated in the new species referred to as *Carnivore chaphamaparvovirus 1* (CaChPV-1), sharing an overall aa identity of 98.6–99.8% in the NS1 protein [4] (Table 1). Recently, a new feline ChPV strain was recovered from feline feces during an outbreak of vomiting and diarrhea in a multi-facility feline shelter in Canada using a viral metagenomic approach [19]. NS1 protein shared 76.0–77.0% aa identity with CaChPV-1 strains previously detected in feline and canine samples, so that this strain was allocated in another species named *Fechavirus* [4,19] (Table 1).

In a recent case–control study carried out in Italy, FeChPV has been identified from fecal and respiratory samples of cats, displaying a correlation with acute gastroenteritis [110]. Since the Italian strains shared more than 97.7% aa identity in the NS1 protein with Canadian prototype viruses, they have been clustered into a monophyletic, well-distinguished species (*Fechavirus*) with respect to the FeChPV strains found in China and currently classified as the species *Carnivore chaphamaparvovirus 1* [2,18,19,110]. *Feciavirus* infection has been correlated with acute gastroenteritis, whereas no correlation has been found with upper respiratory tract disease [110].

Interestingly, in the first report on the detection of ChPV-related parvovirus in domestic cats in China, there was an attempt to isolate the virus on cell cultures, with the observation of a cytopathic effect only up to the fifth generation of cultured cells that were coinfected by FeChPV and FPV [18]. All subsequent studies were mainly focused on the detection of ChPV DNA in cats using molecular approaches, aiming to highlight the epidemiology and genetic heterogeneity of these viruses and to find possible correlations between the presence of viral DNA and clinical signs. To date, however, the information regarding ChPV in cats is still limited and further studies are needed to investigate important aspects such as the host range in vitro (the ability of these viruses to adapt to the in vitro growth is still unclear) and the possibility of detection by immunohistology in the tissues of infected cats [18,19,110].

The potential clinical impact of ChPVs on feline health and their possible role as primary enteric/respiratory pathogens remain to be clarified [19,110]. Indeed, ChPVs DNA was often co-detected in cats with other viral pathogens such as FBoV (Li Y. et al., 2020), feline coronavirus, kobuvirus, and norovirus [110]. Further epidemiological data collected from independent studies in other geographical areas are required to confirm the preliminary findings available so far and assess whether these new parvoviruses can be considered a stable and common component of the host virome or, on the contrary, they play a role in the development of disease in infected cats.

## 5. Conclusions

In the last twenty years, using new molecular techniques and metagenomic approaches for the screening of feline samples, several lineages and species of parvoviruses have been found in association with enteric and/or respiratory disease in cats. Although several aspects concerning epidemiology and virus–host interaction remain to be clarified, some pieces of evidence suggest these emerging feline parvoviruses may act as primary causative pathogens or synergistic agents in the occurrence of clinical signs in cats. Each emerging parvovirus should be included systematically in diagnostic algorithms for detection of feline viral pathogens, chiefly for cats with enteric and/or respiratory disease. Moreover, large structured epidemiological studies and experimental infections might help clarify any possible association of emerging parvoviruses with the occurrence of disease and their distribution in the feline population.

Interestingly, the multi-species circulation of many of these emerging parvoviruses could represent a concrete problem when devising prophylactic measures in animals living in the same environment and, in particular, in mixed cat–dog shelters and veterinary clinics.

Vaccines are not available for emerging parvoviruses of cats, so that vaccination protocols cannot prevent the spread of these viruses, for some of which the cat could represent the host reservoir. Therefore, in order to limit inter- and intra-species spread as much as possible, prophylaxis plans should consider strong disinfection protocols and physical separation, particularly in those facilities housing both dogs and cats.

## Figures and Tables

**Figure 1 viruses-13-01077-f001:**
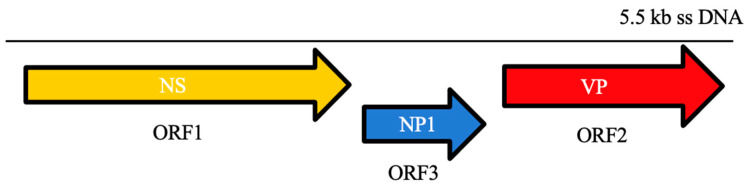
Schematic genome organization of bocaviruses.

**Table 1 viruses-13-01077-t001:** Emerging parvoviruses detected in cats and their current classification.

Subfamily	Genus	Species	Common Names	Country	Year	Detection Source	Reference
*Parvovirinae*	*Protoparvovirus*	*Carnivore* *protoparvovirus* *1*	Canine parvovirus 2 (CPV-2 and its variant 2a, 2b, 2c)	Japan	1993	Stool, blood	[13]
		*Carnivore* *protoparvovirus**2* *	Feline bufavirus (FeBuV)	Italy	2017	Stool, respiratory samples	[14]
	*Bocaparvovirus*	*Carnivore* *bocaparvovirus* *3*	Feline bocavirus (FBoV-1)	Hong Kong	2012	Stool, urine, kidney, blood, respiratory samples	[15]
		*Carnivore* *bocaparvovirus* *4*	Feline bocavirus 2 (FBoV-2)	Portugal	212	Stool	[16]
		*Carnivore* *bocaparvovirus* *5*	Feline bocavirus 3 (FBoV-3)	USA	2014	Stool	[17]
*Hamaparvovirinae*	*Chaphamaparvovirus*	*Carnivore* *chaphamaparvovirus 1*	Feline chaphamaparvovirus (FeChPV)	China	2020	Stool	[18]
		*Fechavirus **	Fechavirus	Canada	2020	Stool	[19]

* Tentatively proposed species.

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
