# Peer review of "Emerging Parvoviruses in Domestic Cats"

_viruses, 2021, doi:10.3390/v13061077_

Round 1
Reviewer 1 Report
Dear Editor,
the authors in an article entitled "Emerging Parvoviruses and Domestic Cats" collected data from the literature and illustrated the presence of different types of parvoviruses in cats. The names are updated according to the ICTV classification. In the chapter Chaphamaparvovirus, it should be clearly stated in which cases ChPV in cats has been detected only by molecular methods and in which it has also been confirmed by classical virological methods.
In this paper, the authors extensively process data from the literature on parvoviruses found in cats. In the introduction, they prepare a systematic review of the family Parvoviridae. Each statement is supported by relevant literature. I suggest that the Introduction begin with the classification of parvoviruses outlined in lines 46-52.
The authors also collect data on the described phenomena of CPV in cats and confirm the statements with the literature published so far. I find that they have cited relevant literature. Given the recent changes in the classification of viruses by ICTV, this review article is very useful and welcome. Authors should be strict in applying the new nomenclature.
L190: According to the new nomenclature, FBoV-1 belongs to the Carnivore bocaparvovirus 3 species, so it should be stated that FBoV-1 is a subspecies.
L220: Chaphamaparvovirus. Caution should be exercised in interpreting results suggesting the presence of ChPV in cats. There is no answer as to whether only DNA of the ChPV virus was detected in cat samples or whether the CPV found in cats was also able to be propagated on cell culture or to be detected by immunohistology in the tissues of potentially infected cats. A clear answer needs to be given here
Author Response
Comments from Reviewer 1 (R.1)
R.1.1: the authors in an article entitled "Emerging Parvoviruses and Domestic Cats" collected data from the literature and illustrated the presence of different types of parvoviruses in cats. The names are updated according to the ICTV classification. In the chapter Chaphamaparvovirus, it should be clearly stated in which cases ChPV in cats has been detected only by molecular methods and in which it has also been confirmed by classical virological methods.
Reply to R.1.1: we agree with the observation of Referee 1. Following the suggestion of the referee, in the chapter Chaphamaparvovirus of the revised manuscript, we have inserted a new paragraph (lines 309-338).
R.1.2: In this paper, the authors extensively process data from the literature on parvoviruses found in cats. In the introduction, they prepare a systematic review of the family Parvoviridae. Each statement is supported by relevant literature. I suggest that the Introduction begin with the classification of parvoviruses outlined in lines 46-52.
Reply to R.1.2: in agreement with Referee 1, in the introduction section of the revised manuscript, the classification of parvoviruses outlined in lines 46-52, was moved to the next paragraph on the morphological description of the virus (lines 23-29).
R.1.3: The authors also collect data on the described phenomena of CPV in cats and confirm the statements with the literature published so far. I find that they have cited relevant literature. Given the recent changes in the classification of viruses by ICTV, this review article is very useful and welcome. Authors should be strict in applying the new nomenclature.
L190: According to the new nomenclature, FBoV-1 belongs to the Carnivore bocaparvovirus 3 species, so it should be stated that FBoV-1 is a subspecies.
Reply to R.1.3: We agree with the observation of Referee 1. Following the suggestion of the referee, we have specified what is required by Referee 1 (lines 230-231).
R.1.4: L220: Chaphamaparvovirus. Caution should be exercised in interpreting results suggesting the presence of ChPV in cats. There is no answer as to whether only DNA of the ChPV virus was detected in cat samples or whether the CPV found in cats was also able to be propagated on cell culture or to be detected by immunohistology in the tissues of potentially infected cats. A clear answer needs to be given here
Reply to R.1.4: we agree with the observation of Referee 1. Following the suggestion of the referee, in the chapter Chaphamaparvovirus of the revised manuscript, we have inserted a new paragraph (lines 309-338).
Reviewer 2 Report
This is an excellent review on parvoviruses in domestic cats. It is obvious that a meticulous search on literature has been made and the outcome has been presented in detail.
My comments are really minor. An inconsistency only was observed in the abbreviation used regarding FPV, the virus, FLP the disease and FLPV probably again the virus.
Moreover, I would suggest including tables and/or figures to summarize and/or highlight the main points of the manuscript.
Author Response
Comments from Reviewer 2 (R.2)
General Comments: This is an excellent review on parvoviruses in domestic cats. It is obvious that a meticulous search on literature has been made and the outcome has been presented in detail.
R.2.1: My comments are really minor. An inconsistency only was observed in the abbreviation used regarding FPV, the virus, FLP the disease and FLPV probably again the virus.
Reply to R.2.1: in agreement with Referee 2, we have changed to "FPV" the abbreviation used for feline panleukopenia virus (lines 114 and 118).
R2.2: Moreover, I would suggest including tables and/or figures to summarize and/or highlight the main points of the manuscript.
Reply to R.2.2: Following the suggestion of the Referee 2, we have added a table (Table 1) to summarize and highlight the main points of the manuscript and a figure (Figure 1) of schematic genome organization of Bocaparvovirus to highlight the presence of ORF3 in the genome, a unique case in parvoviruses.